# Chitosan-Hydroxyapatite Bio-Based Composite in Film Form: Synthesis and Application in Wastewater

**DOI:** 10.3390/polym14204265

**Published:** 2022-10-11

**Authors:** Noureddine Akartasse, Khalil Azzaoui, Elmiloud Mejdoubi, Lhaj Lahcen Elansari, Belkhir Hammouti, Mohamed Siaj, Shehdeh Jodeh, Ghadir Hanbali, Rinad Hamed, Larbi Rhazi

**Affiliations:** 1Laboratory of Applied Chemistry and Environment LCAE, Faculty of Sciences, First Mohammed University, Oujda 60 000, Morocco; 2Department of Chemistry and Biochemistry, Université Du Québec à Montréal, Montréal, QC H3C 3P8, Canada; 3Department of Chemistry, An-Najah National University, Nablus P.O. Box 7, Palestine; 4Institut Polytechnique UniLaSalle Transformations & Agro-Resources Research Unit (ULR7519), 19 Rue Pierre Waguet, BP 30313, 60026 Beauvais, France

**Keywords:** hydroxyapatite, chitosan, biodegradability, biocompatibility, composite, adsorption, antibacterial, antifungal

## Abstract

Water purification from toxic metals was the main objective of this work. A composite in film form was prepared from the biomaterials hydroxyapatite, chitosan and glycerol using the dissolution/recrystallization method. A nanoparticle-based film with a homogenous and smooth surface was produced. The results of total reflectance infrared spectroscopy (ATR-FTIR) and thermal gravimetric analysis (TGA/DTA) demonstrated the presence of a substantial physical force between composite components. The composite was tested for its ability to absorb Cd^2+^ and Zn^2+^ ions from aqueous solutions. Cd^2+^ and Zn^2+^ adsorption mechanisms are fit using the Langmuir model and the pseudo-second-order model. Thermodynamic parameters indicated that Cd^2+^ and Zn^2+^ ion adsorption onto the composite surface is spontaneous and preferred at neutral pH and temperatures somewhat higher than room temperature. The adsorption studies showed that the maximum adsorption capacity of the HAp/CTs bio-composite membrane for Cd^2+^ and Zn^2+^ ions was in the order of cadmium (120 mg/g) > Zinc (90 mg/g) at an equilibrium time of 20 min and a temperature of 25 °C. The results obtained on the physico-chemical properties of nanocomposite membranes and their sorption capacities offer promising potential for industrial and biological activities.

## 1. Introduction

Water of high quality is critical for the survival of all living things on Earth. However, the rapid growth of industry generates large amounts of contaminated waste, which continuously affects water quality. Pollutants present in industrial waste streams are becoming a real threat to nature, so their removal has become a priority. The basic requirements for water purification is a material with high efficiency for toxic organic materials and/or metals, and is natural-based, safe, biodegradable and recyclable.

In this regard, composites may be the ideal material for designing an engineered structure capable of effectively removing toxic elements from wastewater. Biocomposites were used because of their properties such as polarity, density, porosity, stability, selectivity, mechanical strength and dispersibility of composites.

For this reason, the use of biocomposites as a support for heavy metals and organic pollutants removal from water has become the subject matter of several investigations [1,2,3,4]. Some of the most hazardous metals that contaminate water and require immediate attention are chromium, cadmium, mercury, arsenic and lead [5]. These metals have a tendency to accumulate in the tissues of living beings, causing significant health problems and, finally, death. Cadmium (II) for instance is a highly toxic and carcinogenic metal [6]. Several studies show Cd^2+^ accumulates in the food chain, human body and in the environment [7,8,9]. Numerous studies showed cadmium could also target the bones and kidneys of living creatures [10,11].

Several biopolymer-based materials and composites for water treatment are reported in the literature, among these are cellulose [12]-based and lignin-based materials [13] and palm residues [14].

Hydroxy apatite (HAp) is considered a natural material that is useful for many purposes including metal removal from water [15]. Hydroxy apatite’s chemical formula is Ca_10_(PO_4_)_6_(OH)_2_ and it crystallizes in the hexagonal space group system P3/6 m [16]. There are two kinds of cation sites in the form of Ca(I) and Ca(II). Four Ca atoms occupy the Ca(I) position in two layers located at the 0 and 1/2 levels of the mesh. The remaining six occupy the Ca(II) position in two equilateral triangles at the 1/4 and 3/4 levels in two equilateral triangles. The six PO_4_ tetrahedral ions are the backbone of the hexagonal mesh. The PO_4_ ion assemblies are in the form of a honeycomb, which constitutes the reinforcement of the network and provides a great stability to the structure of the HAp. This assembly is parallel to the axis “c”, which leads to open tunnels. The two OH^-^ ions are found approximately in the tunnels parallel to the “c” axis. These tunnels are extremely significant in the physicochemical properties of HAp. Due to the existence of these tunnels, HAp can behave as ion exchangers in which different ions can be substituted [17]. Different methods of preparing HAp were described in the literature, such as the neutralization method [18] and the double decomposition method [17].

In our laboratory, our research mainly concentrates on improving the physicochemical and morphological properties of the HAp by constructing composites based on hydroxyapatite and natural polymers such as chitosan, using a new preparation method known as dissolution precipitation.

Chitosan is an N-deacetylated chitin derivative (2-acetamido-2-deoxy—D-glucose via a (1–4) linkage), which exists mainly in the cell walls of certain fungal microorganisms [19,20]. At low pH, chitosan protonates the amino group, resulting in a polycationic species. Chitosan’s cationic property can be reversed by sulfonation to add an anionic character.

Chitosan (CTs) is a non-porous semi-crystalline substance [19] that is very stable in the solid state. Its chemical activity is attributed to the free amino group which gives it a high positive charge density [21], and also allows for adding a new functionality by acetylation or alkylation [21]. Chitosan is biodegradable [19,22,23] and non-toxic [20]. Chitosan is a crystalline polymer similar to cellulose and it can be produced in fiber form. It has unique properties that distinguish it from other natural polymers such as its high ability for water retention, metal complexation, adsorption of organic molecules, ion chelation [20,24] and adhesion to negatively charged surfaces [25].

It has found widespread use in medicinal sectors, the textile industry, medication delivery and environmental protection [26,27]. We have also demonstrated in recent years that the presence of specific metal ions in bodies of water is a high priority environmental concern. One big disadvantage is that, unlike environmental compounds, which are more susceptible to biological decay, metal ions do not dissolve on their own and thus pose the same risk to the atmosphere and human health. Mn, Cr, Zn, Fe and Cd are metals found in almost 85 percent of industrial wastewater [25].

Both chitosan and HAp have the functionality required for strong interaction and compatibility represented by H-bonding. The main objective of this article is to prepare a novel composite of Chitosan/Hydroxyapatite HAp/CTs in film form. A new process for making a composite film is offered in the work. The developed film is in a membrane form that is flexible and transparent. The film can perform a dual function in water purification, It exhibited a high affinity for a variety of toxic metals as well as antibacterial activity against both Gram-negative and Gram-positive bacteria. FT-IR, DSC, XRD, TGA, and SEM studies were used to characterize the produced film.

## 2. Materials and Methods

### 2.1. Materials

Chitosan (CTs) was acquired from Aldrich Chemical Company, Inc. (Milwaukee, WI, USA) and Hydroxyapatite (HAp) was produced according to a previously published technique [17].

Ammonium hydrogen phosphate (NH_4_)_2_HPO_4_ (99%), calcium nitrate Ca(NO_3_)_2_·4H_2_O (99%), lysine, Acetic acid (99%), and cadmium chloride (CaCl_2_) were all purchased from Sigma Aldrich (Waltham, MA, USA) and utilized precisely as received, with no additional purification. Both runs utilized distilled water of high quality.

A Cd^2+^ and Zn^2+^ stock solution with a concentration of 1200 ppm (S0) was prepared.

Then, a series of diluted Cd^2+^ and Zn^2+^ solutions with concentrations of 300, 210, 170, 120, 100, 80, and 40 ppm were produced.

### 2.2. Methods

To collect IR spectra, a Nicolet 6700 Fourier transform infrared (FTIR) spectrometer equipped with the Smart SplitPea micro-ATR adaptor was employed. Thermo Electron’s OMNIC software (Thermo Fisher Scientific, Waltham, MA, USA) was utilized to process spectral data. The SplitPea is a clever horizontal attenuated absolute reflectance micro sampling attachment for Thermo Electrons Nicolet FT-IR spectrometers. The SplitPea is intended for ATR testing of very small samples of solids (powder or film) as well as liquids. The piece is encrusted with a diamond ATR crystal. The following parameters were employed: The resolution is 4 cm^−1^, the spectral range is 600–4000 cm^−1^, and the number of scans is 64. Emission-scanning electron microscopy was performed using an SU 8020, 3.0 KV SE (U) (SEM). Specimens were frozen in liquid nitrogen before being shattered, assembled, gold/palladium coated, and examined with a 10 kV applied stress. The composites were thermogravimetrically analyzed using the TGA Q500 and Q50 TA equipment at a heating rate of 10 °C/min and temperatures ranging from 20 to 900 °C.

Metal cations concentrations were determined using an atomic Absorption Spectrophotometer, AAS (a Varian A.A.400 spectrometer).

An XRD (XPERT-PRO, PW3050/60) investigation was performed at room temperature using a Diffractometer with CuK radiation (1.5418) within the 2*θ* ranging from 20° to 80° with a sweep rate of 2°/min. The phase of the reaction product was analyzed using a TOC/TN Analyzer multi N/C 2100/2100 because the reaction products were primarily soluble in the aqueous process. Total carbon (TC) was used rather than total organic carbon (TOC) to characterize the carbon content of the aqueous process since this value contains all carbon (organic and inorganic) in the aqueous effluent.

### 2.3. Antimicrobial Evaluation of Composites Materials

The antibacterial activity of the generated composites was evaluated using the WHO-recommended disk diffusion methodology and susceptibility test method, as well as the French standard NF-U-47-107 AFNOR 2004.

Sterile distilled water, Muller-Hinton Broth (Biokar, Darmstadt, Germany), Potato Dextrose Agar (PDA), Muller-Hinton Agar (Biokar), sterile paper discs; Test Tubes; Petri plates 90 mm were the reagents required for the antimicrobial tests. The microorganisms utilized in this investigation included Gram-negative *E. coli* and two-gram positive *Micrococcus luteus* and *Bacillus subtilis*, in addition to Candida (*Candida albicans fungus*).

Bauer et al. [28] described the disc diffusion approach for antimicrobial susceptibility, testing a bacterium colony (tuned to the 0.5 McFarland standard), which was used to mow Muller Hinton agar plates evenly using a sterile wipe. The plates were used for sensitivity test after drying for 15 min. The discs were mounted on the Mueller-Hinton agar surface after being impregnated with a variety of plant extracts. Each test plate consists of six disks. There were four treated discs and one positive control (a normal generic antibiotic disc containing 1.0 mg/mL tetracycline). Aside from the controls, each plate had four treated discs mounted approximately equidistantly apart. Based on the form of bacteria tested, the plate was then incubated *E. coli* and *Micrococcus luteus* and incubated at 37 °C for 24 h, whereas Bacillus subtilis was incubated at 33 °C. Candida albicans was incubated for 48 h at 37 °C with cycloheximide as an antifungal [29]. After incubation, the plates were examined for the inhibitory zone, which was measured with calipers. The method was done three times to verify dependability.

### 2.4. Adsorption

Cadmium (II) and Zinc (II) were chosen for this research. The batch technique was employed to carry out the adsorption process [30,31,32]. This method used 10.0 mg of adsorbent per 10 mL of metal solution with concentrations ranging from 40 to 300 mg/L, and the adsorption was performed at 25 °C. The impact of adsorption duration and pH was examined, and the pH was changed by adding either HNO_3_ or NH_4_OH. The nature of the adsorption process was investigated using thermodynamic parameters [31,32,33,34].

AAS flame atomic absorption spectroscopy was used to determine the change in metal concentration. The capacity for adsorption of metal ions by composites was calculated as shown in Equations (1) and (2) [35]:(1)$$\%\mathrm{removal}=\frac{C_{0}-C_{e}}{C_{0}}\cdot 100$$
(2)$$Q_{e}=\frac{C_{0}-C_{e}}{W}V$$
where *C*_0_ and *C_e_* are the initial and equilibrium Cd^2+^ concentrations (ppm), *Q_e_* is the equilibrium adsorption capacity (ppm), *W* is the adsorbent weight (mg), and *V* is the solution volume (L).

#### 2.4.1. Adsorption Isotherm

The Langmuir, Freundlich and Redlich Peterson isotherm models were employed to assess the adsorption behavior of cadmium molecules on the HAp/CTs composite surface. [31]. The Langmuir isotherm model is presented in Equations (3) and (4):(3)$$\frac{\;C_{e}}{Q_{e}}=\frac{1}{q_{max}}C_{e}+\frac{1}{q_{max}K_{L}}$$
where *C_e_* is the equilibrium concentration of Cd^2+^ (ppm), *Q_e_* is the amount of adsorbate adsorbed per unit mass of HAp/CTs composite at equilibrium (mg/g), *q_max_* is the maximal monolayer adsorption capacity of the adsor-bent (mg/g), and *K_L_* is the Langmuir constant (L/mg).

In the Langmuir isotherm model, the dimensionless constant separation factor indicated in Equation (4) may be used to forecast whether adsorption is favorable or unfavorable.
(4)$$R_{L}=\frac{1}{1+K_{L\;}C_{e}}$$
where *K_L_* denotes the Langmuir constants and *C*_0_ denotes the initial Cd^2+^ concentration. Adsorption is considered unfavorable if the R_L_ value is greater than one, favorable if it is between one and zero and linear if it is one.

The non-ideal adsorption process with heterogeneous surface energy is represented by the Freundlich isotherm (Equations (5) and (6)).
(5)$$\ln\left(q_{e}\right)=\ln k_{f}+\frac{1}{n}\ln C_{e}$$
*Q_e_* = *K**_F_ C_e_*1/*n*(6)where *K_F_* is constant and indicates the relative adsorption capacity and 1/*n* is the adsorption intensity [35].

If 1/*n* between 0.1 and 0.5, the adsorption is favorable; if 1/*n* is greater than 2, the adsorption is unfavorable.

#### 2.4.2. Adsorption Kinetics

The adsorption rates of Cd^2+^ on the HAp/CTs surface were investigated using pseudo-first-order and pseudo-second-order kinetic models. The linearized versions of the rate equations [31] were computed using Equations (7)–(10) [35]:(7)$$\ln(q_{e\;}-q_{t})=\ln q_{e}-K_{1}t$$
(8)$$\frac{t}{q_{t}}=\frac{1}{K_{2}q_{e}^{2}}+\frac{t}{q_{e}}\;$$*Q_t_* = *K*_id_*t*^1/2^ + *Z*(9)
(10)$$\ln\frac{K\left(T_{2\;}\right)}{K\left(T_{1}\right)}=\frac{Ea}{R}\cdot \left(\frac{1}{T_{1}}-\frac{1}{T_{2}}\right)$$
ln(1 − *F*) = −*K*_fd_**t*(11)where *q_e_* is the equilibrium adsorption capacity (mg/g) and *Q_t_* is the temperature-dependent adsorption capacity. The pseudo-first-order rate constant (min^−1^) is denoted by *K*_1_, and the pseudo-second-order rate constant (g/mg·min) is denoted by *K*_2_. *K*_id_ is the intra-particle diffusion rate constant (mg g^−1^ min^−1/2^), and Z can be used to calculate the boundary layer thickness (mg/g).

In Equation (11), *F* represents the fractional achievement of equilibrium (*F* = *q_t_*/*q_e_*), and *K*_fd_ (min^−1^) represents the rate of film diffusion.

The Gibbs energy (*G*°), enthalpy (*H*°) and entropy (*S*°) were calculated using the formulae indicated below (Equations (12)–(14))
*K_c_* = *C_ads_*/*C_e_*(12)
Δ*G*° = −*RT*ln*K_c_*(13)
(14)$$\ln K_{s}=\frac{ƊS}{R}-\frac{ƊH}{RT}$$
where *K_c_* is a thermodynamic constant, *C_ads_* is the equilibrium Cd^2+^ adsorbed quantity (mg/L), *C_e_* is the equilibrium concentration (mg/L), *T* is the solution temperature (*K*) and *R* is the ideal gas constant (J/mol·K) [36].

### 2.5. Synthesis of HAp/Chitosan Film

The dissolution/recrystallization process was used to create the HAp/CTs biocomposites [17,37,38,39]. In this procedure, a chitosan solution in 5% aqueous acetic acid was prepared. A separate solution of HAp was prepared by dissolving chitosan in a 5% aqueous solution of acetic acid [22,23,40,41]. The two solutions were then mixed with stirring at 45 °C for 2 h. The resulting solution’s pH was then adjusted to 10.5 using ammonia and stirring was maintained at 60 °C for another 12 h before being decanted onto a petri plate and left at room temperature for 48 h, and a composite in the form of a transparent film was produced. For glycerol, 5% of the total composite weight was used as plasticizer to enhance membrane smoothness, flexibility and transparency. The obtained membrane was washed thoroughly with ethanol. Four composites with various compositions ratio by weight of HAp/CTs: HAp/CTs 30/70, HAp/CTs 75/25, HAp/CTs 80/20, and HAp/CTs 85/15 were developed.

The stability of the composite in aqueous solutions at various pH values (3.0 to 10) for 2 h was evaluated. The change in the mass of the composite before and after immersion in an aqueous solution and the swelling behavior were monitored. The composite showed no change in weight and very low water swelling after immersion for about 2 h.

### 2.6. Surface Characterization

Atomic Force Microscopy (AFM) was utilized for surface analysis to examine the changes in surface morphology of mild steel at 303 K after 6 h of immersion.

The AFM measurements were taken with a VEECO (Oyster Bay, NY, USA) CPII atomic force microscope (MPP-11123) at a resonance frequency of 335–363 kHz with spring constants k of 20–80 N/m at 5 m. The X-ray photoelectron spectroscopy (XPS) spectra were obtained using a Physical Electronic (Chanhassen, MN, USA) PHI 5700 spectrometer and a non-monochromatic Mg-K radiation source (300 W, 15 kV and 1253.6 eV). A hemispherical multichannel detector operating in the constant pass energy mode at 25.9 eV was used to capture spectra from a 720 m diameter study area. PHI ACESS ESCA-V6 was utilized to analyze the acquired X-ray photoelectron spectra. The software was processed using the MultiPak 8.2 B kit. The binding energy data were compared to the C1s signal from incidental carbon (284.8 eV). A Shirley-type backdrop and Gauss-Lorentz curves were used to compute the binding energies.

## 3. Results and Discussion

### 3.1. HAp/CTs Characterization

#### 3.1.1. FT-IR Spectra of HAp/CTs Composites and Starting Materials

Figure 1 depicts an overlay of the IR spectra of the four produced composites.

The absorption bands of the PO_4_^3−^ ions that comprise the apatitic network are distinguished by two absorption domains (the 1100–900 cm^−1^ domain and the 600–500 cm^−1^ domain, respectively). The hydroxyl ion of hydroxyapatite is responsible for the absorption band at 3570 cm^−1^ [17]. The NH_2_ and OH groups of chitosan are responsible for the broadband at 3449 cm^−1^. The hydroxyapatite/chitosan compound shifts this band to 3566 cm^−1^ [19]. The vibrations of N-H and O-H of chitosan are represented by the bandwidth between 3100 and 3500 cm^−1^ [42]. The C-H stretch vibration of the methylene groups is responsible for the band at 2926 cm^−1^. The deformation vibration of N-H of the NH_2_ groups is represented by the band at 1588 cm^−1^ [2]. The C-OH and C-N groups’ stretching vibrations correspond to absorption bands at 1076 and 1379 cm^−1^, respectively.

It can be seen that all the distinctive bands of HAp and CTs appear in the spectra of the hydroxyapatite/chitosan composite, except for a slight displacement of the specific bands of the amine groups, which have changed to lower wavenumbers (3430 cm^−1^), indicating the possibility of interactions between hydroxy-apatite and chitosan (NH_2_ and Ca^2+^).

#### 3.1.2. X-ray Diffraction

Figure 2 illustrates the XRD spectra of the four composites. Before XRD analysis, the composites were calcinated at 900 °C. All spectra show two intense peaks around 26° and 31.8°, which represent the crystalline regions (002) and (211), respectively.

The XRD spectra show that crystallinity is proportioned to the organic contents of the composites. As the percentage of the organic fraction in the composite increases (a and b), the crystallinity decreases. For instance, Sample c clearly shows the HAp lattice planes at (200), (111), (300), (301), (131), and so on, then the intensity of the peaks decreases and broaden in sample b. The splitting of broadened peaks in the two angular area of approximately 31–34° of composites (a) and (b) indicates a lower degree of crystallinity. The results supported the FT-IR results, indicating the presence of a direct interaction between the composite components.

The characteristic lines at 25.8° (002) and 39.6° (310) are familiar to compute the crystal size of the apatitic nanoparticles in the hydroxyapatite/chitosan composite, based on Scherrer’s formula (Table 1).

The results of the calculations show that the size of the apatitic nanoparticles is reduced more in the composite than in the case of pure HAp. Further, the lower the apatite/biopolymer mass ratio in the composite, the smaller the size of the apatite nanoparticles. This could probably be due to the synergy of three factors:−The concentration of cations (Ca^2+^) and anions (PO_4_^3−^) in the reaction medium is the first component to consider. As a result, the lower the concentration of these ions, the more nanoparticles of tiny size and dispersion are produced.−The second factor is the influence of the amount of acetic acid in the chitosan solution, used initially to dissolve it. This is because the greater the amount of chitosan, the more you need an additional amount of acetic acid to dissolve it. Thus, acetic acid can further alter the crystals of hydroxyapatite, when its concentration in the reaction medium is greater. This causes the apatitic nanoparticles to shrink in size.−The third factor is linked to the dispersive power of chitosan, which grows in accordance with the concentration of this biopolymer in the reaction medium. The size of the apatitic nanoparticles thus decreases when the dispersive power of the chitosan increases.

#### 3.1.3. Thermal Analysis

Pure samples of starting materials and composites were subjected to thermogravimetric and differential thermal analysis (TGA/TDA).

A temperature range of room temperature to 1000 °C was used, with a rising rate of 10 °C/min. Thermogravimetric analysis was done to demonstrate the influence of HAp on chitosan thermal stability; the results are presented in Figure 3.

The mass loss was recorded to occur at three stages. The first loss (6.24 wt%) occurred at 150 °C, so it could be related to the desorption water (endothermic transformation). the second loss (11.12 wt%) occurred between 200 and 290 °C and resembles the rapid decomposition of CTs in the composite. At a temperature of over 300 °C, the last loss occurred (17.35%), which could be attributed to the continuous decomposition of chitosan.

The TDA curves for the synthesized HAp demonstrate that water molecules were lost at around 100 °C, which may be attributed to surface evaporation. At high temperatures, two exothermic peaks between 250–400 °C and 550–750 °C were observed, and the peaks are characteristics of the combustion of the organic material contained in the composite.

#### 3.1.4. SEM Analysis

SEM (scanning electron microscopy) was used to investigate the influence of chitosan-HAp interaction on composite morphology, as shown in Figure 4. SEM micrographs show that the dense structure of the composite demonstrates the improved mechanical properties of the generated film.

Scanning electron microscopy (Figure 4) and atomic force microscopy (Figure 5) show apatitic nanoparticles in the form of needles. The dispersion of the particles is not completely homogeneous as indicated by scanning electron microscopy.

#### 3.1.5. Atomic Force Microscopy

Atomic force microscopy was utilized to visualize and quantify how the tested inhibitor reduces corrosion rates by evaluating surface roughness with and without HAp/CTs.

AFM micrographs of a metal surface without and with 1 g/L HAp/CTs are shown in Figure 5. In the absence of HAp/CTs, the mild steel composition is more corroded (Figure 4), with an average roughness of 1.3 μm. In the presence of HAp/CTs at the optimum concentration (1 g/L), the average roughness was decreased to 500 nm, owing mostly to the formation of a protective layer [24].

#### 3.1.6. X-ray Photoelectron Spectroscopy (XPS)

Using high-resolution XPS, the figures from X-ray photoelectron spectroscopy (XPS) on composites based on hydroxyapatite and chitosan in the form of a membrane may be split into four separate components (Figure 6a).

The composite particles in the current investigation were created utilizing two distinct ways. First, the surface composition of the three samples was investigated using X-ray photoelectron spectroscopy (XPS). The binding energy of N1S is reflected by a prominent peak at 399.3 eV in the XPS spectra of all samples shown in Figure 6. The nitrogen signal, which only originates from chitosan’s amino groups, demonstrates the presence of chitosan in the biocomposites surface layer and implies that chitosan molecules are genuinely interlocked, making true nanocomposites. At 287.8, 286.4 and 284.9 eV, the C1 Speak is divided into three pieces. The peaks represent the O-C-O, C-OH and C-C bonds found in chitosan (Figure 6a). At 532.8 eV, a third peak appears, which corresponds to the O1 ascribed to the hydroxyl groups (Figure 6b).

#### 3.1.7. Schematic Model of the Composite

Figure 7a illustrates a schematic model of the composite HAp/CTs, which might be utilized to explain the data obtained by ATR-FTIR, TGA/TDA and SEM, all of which imply a direct connection between Hap and CTs. The diagrams show the locations of interaction between HAp and CTs. The diagram depicts the formation of a link between the NH_2_ groups of CTs and the Ca of hydroxyapatite. The mechanism that led to the formation of H-bonding could be explained by the method of making the composites, which consisted of four stages, the first of which was the dissolution of the reagents in an aqueous medium, followed by the diffusion of water through the two matrices, organic and inorganic, and the creation of links between the functionalities of the two components begins, leading to the formation of HAp/CTs composites (Figure 7a). A photo of the created film is given in Figure 7b; the photo reveals a clean smooth film was obtained, which is another indicator of the materials’ strong computability.

#### 3.1.8. Antibacterial and Antifungal Test

Antimicrobial characteristics of HAp/CTs composites were studied.

The test results demonstrated that the degree of inhibition is determined by the composite factor ratios [35]. Figure 8 summarizes the findings. Composite HAp/CTs 20/80 inhibited the development of *E. coli* and *B. subtilis* (B.S.) with inhibition diameters of 12.0 and 13.5 mm, respectively, and demonstrated a significant inhibition of M. Luteus with a diameter 14.0 mm.

In addition to antifungal activity, composite HAp/CTs 15/85 inhibited Candida Albicans with a diameter of inhibition of around 8.0 mm. In the positive control, the diameter of inhibition for *M. Luteus* was approximately 25.0 mm, 26.0 mm for *E. Coli*, 25.0 mm for *B. Subtilis* and 25.0 mm for Candida.

The other composites (HAp/CTs 30/70, HAp/CTs 25/75 and HAp/CTs 20/80) exhibited no antimicrobial activity, with the exception of composite HAp/CTs 25/75, This showed exclusively anti-B.S. activity with a diameter of inhibition of 14.0 mm.

### 3.2. Adsorption of Metals

#### 3.2.1. Metal Concentration Effect

The impact of Cd^2+^ initial concertation on the rate of adsorption was studied, and the results are shown in Figure 9A. The quantity of Cd^2+^ adsorbed by the substrate (Q_e_) was displayed as a function of the starting concentration of Cd^2+^ in this image (Ce). The adsorption process was carried at 25 °C for two h, time beyond which there is no longer change in the concentration of Cd^2+^. As shown in Figure 9A, the four composites show an almost similar rate of absorbency, and the adsorption capacities of the HAp/CTs composites increased linearly with increasing initial concentrations of Cd^2+^. At 100 mg/L Cd^2+^ concentration, the rate of adsorption for all composites reached the plateau. The HAp/CTs 20/80 composite, with a maximum adsorption capacity of 120 mg/g, provided the best adsorption results. The results suggest the adsorption initially occur though diffusion into the pores of the composites. The complexation process then takes over as a second stage of adsorption, and during this stage, the adsorption by coordination of metal ion amine and hydroxyl groups occurs. The results indicate that the Cd^2+^ sorption would be mainly attributable to surface complexation process, since the diffusion process is limited by pore size.

#### 3.2.2. pH Effect

pH is a vital factor in the process of metal ion adsorption. The variation in the composite surface charge could affect rate of absorbency and process of adsorption. Results obtained from adsorption as a function of pH value are presented in Figure 9B. The results demonstrate that raising the pH value enhances the adsorption capacity of the composites HAp/CTs 15/85, HAp/CTs 20/80 and HAp for Cd^2+^. At a low pH, the composite surface is protonated, causing a repel of positively charge metal ion. In this case, the adsorption occurs only by diffusion into the pores of the composites. However, at a pH of about 7, the amine groups are deprotonated, with lone pair of electrons available for coordination with metal ions causing higher adsorption. At pH > 8, the cadmium ions tend to precipitate as a hydroxide (Cd(OH)^+^ and Cd(OH)_2_). Therefore, the maximum adsorption of Cd^2+^ by the HAp/CTs composites was achieved at pH = 7.

#### 3.2.3. Contact Time Effect

The influence of contact time on Cd^2+^ adsorption by composites was also investigated.

Measurements were taken every 25 min for a total of 120 min. Figure 9C depicts the results. The plateau was reached after 25 min of contact time, as seen in the figure. The adsorption of Cd^2+^ was particularly fast during the first 20 min, which might be attributed to the availability of the active site represented by the hydroxyl and amino groups, as well as the availability of nonbonding electrons on these groups, which promotes a speedy complexation with the metals. The slower diffusion rate via the pores into the interior of the composite may explain the slow adsorption rate at later stages of the process [43].

#### 3.2.4. Adsorption Isotherms

To describe the exact mechanisms involved in the adsorption of Cd^2+,^ the two most generally used models, Langmuir and Freundlich, were applied to the experimental data using Equations (4) and (5) above. The Langmuir model assumes that the adsorbent surface is homogenous and that there are defined adsorption sites with no interactions between adsorbate and absorbent. According to this model, adsorption occurs by the forming of an adsorbate monolayer. Freundlich’s model is built on an empirical equation that represents the difference of adsorption energies with adsorption amount. The heterogeneity of the adsorption sites explains this process. The Freundlich equation, unlike the Langmuir formula, does not have an upper limit for adsorption. Figure 10A,B demonstrates the curves obtained by using the two models, Langmuir and Freundlich.

Freundlich and Langmuir models both demonstrated a linear interaction between the quantity of Cd^2+^ adsorbed in (mg/g) and the initial concentration. Table 2 summarizes the parameters obtained from applying the two models. Based on the correlation coefficient (R^2^) values obtained by the two models, it is possible to conclude that the Langmuir model is better suited to understanding adsorption.

### 3.3. Adsorption Kinetics

The kinetic parameters for Cd^2+^ ion adsorption onto HAp/CTs composites were determined by fitting the pseudo-first-order model (Equation (7)), the pseudo-second-order model (Equation (8)) and intra-particle diffusion (Equation (9)) to the experimental data.

Figure 11a–c depicts the plots produced by the three models. Cadmium and zinc adsorption on HAp/CTs composite were suited to Equation (7) for the pseudo-first-order model, where k_1_ (min1) is the first order constant and q_e_ and q_t_ are the amounts of cadmium and zinc mass absorbed at equilibrium and time t, respectively. According to Figure 11a, the data did not converge well, and the regression factors were 0.914 for Cd^2+^ and 0.921 for Zn^2+^. Moreover, the estimated value of the adsorbed amount at equilibrium (Q_e_ calculated) is far from the experimental value (Q_e,exp_) (see Table 3). This means that pseudo-first-order is not a good model to represent the adsorption study is shown in Figure 11a. The other model which used to represent our study is pseudo second order. Figure 11b shows the results of using Equation (10) and graphing t/qt vs. t. After 15 min, the regression coefficients for Cd^2+^ and Zn^2+^ were 0.985 and 0.991, respectively. The adsorbed amount at equilibrium (Q_e,Calc_) and the experimental (Q_e,exp_) were almost very close. This shows that the pseudo-second order was best choice to represent our adsorption study for both cadmium and zinc. Another important study was carried is the intra-particle diffusion and it was represented using Equation (10). Plot of qt vs. t^0.5^. This model could reflect the adsorption process analysis if the data in Figure 11c converges and passes through the origin. As notedthe model was used to match the experimental data of cadmium and zinc adsorption by HAp/CTs, although they did not converge. Moreover, no straight line across the origin was obtained. As seen in Table 3, the only converge occurred after one minute, and the regression coefficient was 0.995. This means that the process occurred in two stages, and although this model is not the rate-determining phase, it could be a part of the cadmium and zinc adsorption mechanism by HAp/CTs. Table 4 displays the intra-particle diffusion rate constants. The first phase in Figure 11c was very rapid and normally reflects liquid film diffusion, and it happened within the first minute. The second stage, on the other hand, was very slow and represented the diffusion of cadmium and zinc through the film. This leads to the conclusion that the adsorption process was a mixed process. The liquid film diffusion model, which assumes the transport of cadmium and zinc through a liquid film covering the solid adsorbent, is another kinetic model [40]. The liquid film diffusion model may be expressed using Equation (11). The results did not converge well when the diffusion film model was applied to Cadmium and Zinc adsorption data on HAp/CTs, and the linear regression coefficients were not as expected. The data are represented in Figure 12 and Table 3. A linear plot of ln(1−F) vs. t (Figure 12) with zero intercept demonstrated that the kinetics of the adsorption process were regulated by diffusion through the surrounding liquid layer, according to Equation (11). The experimental adsorption results of metals by HAp/CTs from an aqueous solution at various temperatures, as shown in Figure 11, did not exhibit straight lines that went through the origin and had coefficients of determination of 2.312 and 2.264, respectively, for Cd^2+^ and Zn^2+^ (Table 4). The final conclusion from this study is that the adsorption process followed the pseudo-second-order [23,26]. The obtained correlation coefficients values were very close to 1 (R^2^ = 0.999). These findings are consistent with those seen in the literature, indicating the adsorption of Cd^2+^ on HAp/CTs composite follows a pseudo-second-order type law [44,45,46].

### 3.4. Adsorption Affinity for Cd^2+^ and Zn^2+^

The selectivity of the HAp/CTs composites for metal ions was investigated using solutions of zinc (II) and cadmium (II) with concentrations ranging from 20 to 300 ppm.

Figure 13a depicts the collected results. The data clearly reveal that the membranes have better Cd^2+^ and Zn^2+^ selectivity. The result could be used as proof that surface complexation is primarily responsible for controlling the sorption mechanism rather than the diffusion process. Since zinc (II) has a smaller ionic radius (0.74 Å) than Cd^2+^ (0.94 Å), if the sorption is controlled by the diffusion process, the composite would have more affinity for Zn^2+^.

### 3.5. Thermodynamic Results

Table 3 shows the thermodynamic parameter quantities obtained for the adsorption of Cd^2+^ and Zn^2+^ ions onto HAp/CTs: Gibb’s energy (ΔG°), entropy (ΔS°) and enthalpy (ΔH°).

The slope and intersection of the plot of ln(k_d_) vs. 1/T (Figure 13b) were used to calculate the values of ΔH° and ΔS°, as required by Equation (13). The values of ΔG° were calculated using Equation (12). A negative value for ΔG° indicates that the adsorption is spontaneous.

The value of ΔG° falls as temperature rises, indicating that adsorption is spontaneous for both metal ions Cd^2+^ and Zn^2+^, and the value of the standard enthalpy ΔH° is positive, which shows that the adsorption is endothermic [47,48,49,50,51].

Positive entropy values during adsorption can be ascribed to structural variations in the adsorbents. Such positive results indicate that randomness develops at the solid/solution interface during Cd^2+^ and Zn^2+^ sorption. Positive ΔS° readings also indicate the composites’ attraction for Cd^2+^ and Zn^2+^ metal ions [49]. Table 3 displays the obtained data.

Three possible ionic interactions may be responsible for the adsorption process’s spontaneous nature as shown in Figure 14. The first two are related to the interaction between the Cd^2+^ and Cd^+^ and OH. As shown in Figure 14, Cd has a resonance structure; the resonance structure has an ammonium group. The ammonium group undergoes interaction with the OH ions of the composite.

## 4. Conclusions

A natural-based composite in the form of film was designed and synthesized based on HAp, chitosan, and glycerol. Several composites were synthesized by the dissolution/reprecipitation method using various rations of HAp and chitosan. The composites were evaluated as adsorbents for toxic metals form wastewater. The prepared composites were subjected to analysis by ATR-FTIR, X-ray diffraction, SEM and thermal analysis. The results showed a substantial interaction between composite components.

SEM scans revealed that the composite surface was made up of spherical particles ranging in diameter from 30 to 120 nm. The ability of the composite to extract metals from water was evaluated in a micro-extraction process of Cd^2+^ and Zn^2+^. The absorption investigation results revealed that Cd^2+^ and Zn^2+^ absorption on the composite was spontaneous, the ΔG° value was negative and the ΔS° and ΔH° values were positive. The low ΔH° values imply that only physical forces including van der Waals forces, electrostatic interaction and hydrogen bonding were involved in the adsorption. The adsorption studies showed that the maximum adsorption capacity of the HAp/CTs bio-composite membrane for Cd^2+^ and Zn^2+^ ions was in the order of cadmium (120 mg/g) > Zinc (90 mg/g) at an equilibrium time of 20 min and a temperature of 25 °C. The composite performed better with Cd^2+^ and Zn^2+^ when the Langmuir and Freundlich models were applied to the experimental data, and it was discovered that the Langmuir isotherm model best suited the experimental data for the adsorption of Cd^2+^ onto the HAp/CTs composites. The adsorption process was also explained using several kinetic models. The results show that Cd^2+^ adsorption onto HAp/CTs is better suited to a pseudo-second-order kinetic mechanism. Cd^2+^ ion adsorption on the composites occurred preferentially at neutral pH. These findings imply that the Cd^2+^ ions were most likely adsorbed onto the composites’ uncharged surfaces, such as amine and hydroxyl groups. The HAp/CTs 20/80 composite inhibited Gram-positive and Gram-negative bacterias moderately in an antibacterial and antifungal study.

## Figures and Tables

**Figure 1 polymers-14-04265-f001:**
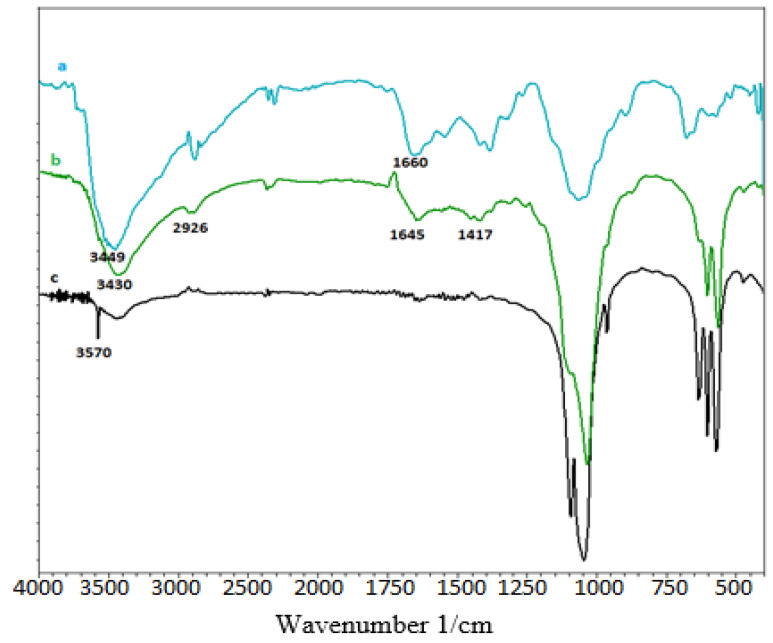
FT-IR of, (**a**): chitosan; (**b**): HAp/CTs; and (**c**): HAp.

**Figure 2 polymers-14-04265-f002:**
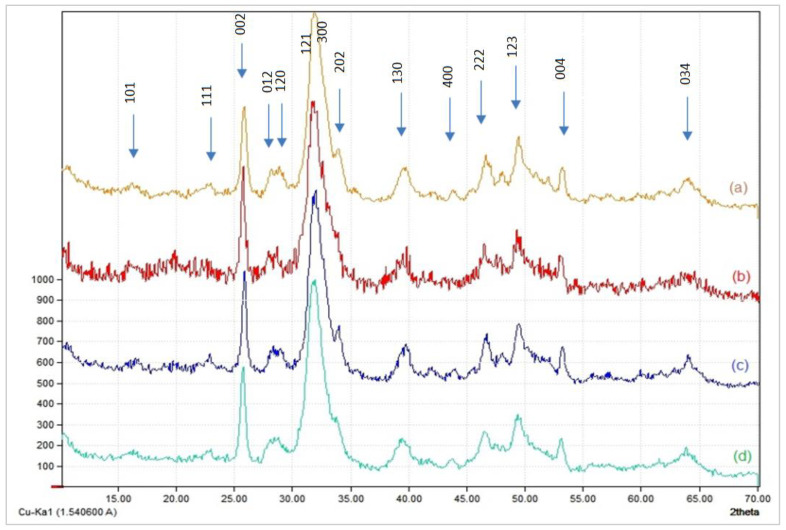
X-ray diffraction patterns of HAp/CTs composites: (**a**): 30/70; (**b**): 25/75; (**c**): 20/80 et; (**d**): 15/85.

**Figure 3 polymers-14-04265-f003:**
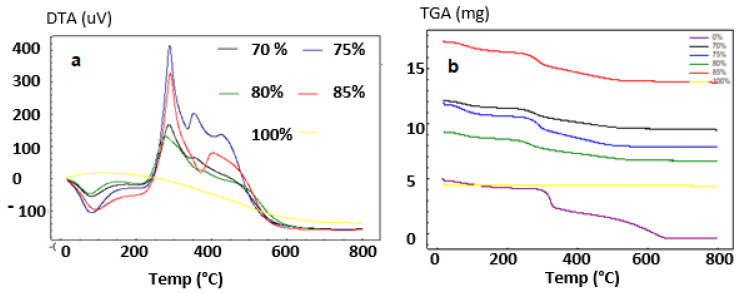
DTA (**a**) and TGA; (**b**) Spectrum of HAp/CTs composites.

**Figure 4 polymers-14-04265-f004:**
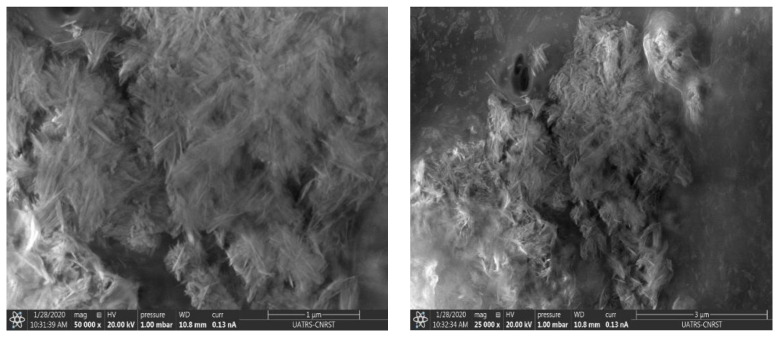
SEM images of HAp/CTs composite.

**Figure 5 polymers-14-04265-f005:**
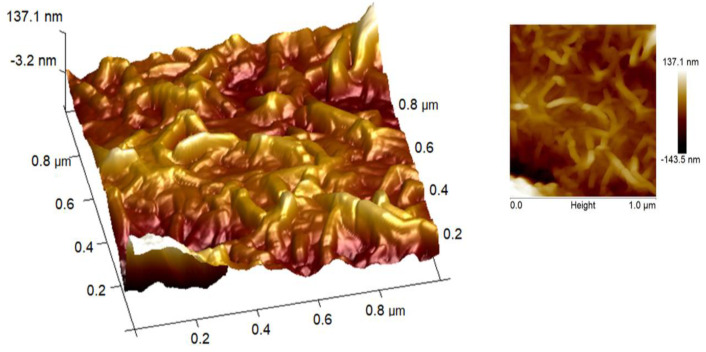
AFM micrographs in HAp/CTs at 1 g/L.

**Figure 6 polymers-14-04265-f006:**
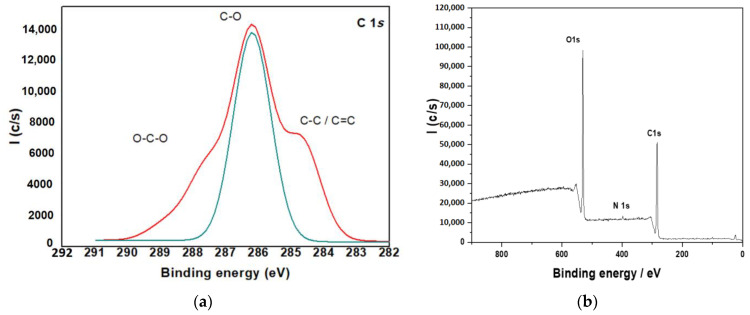
XPS for (**a**) C 1s; (**b**) O 1s core level of CTs.

**Figure 7 polymers-14-04265-f007:**
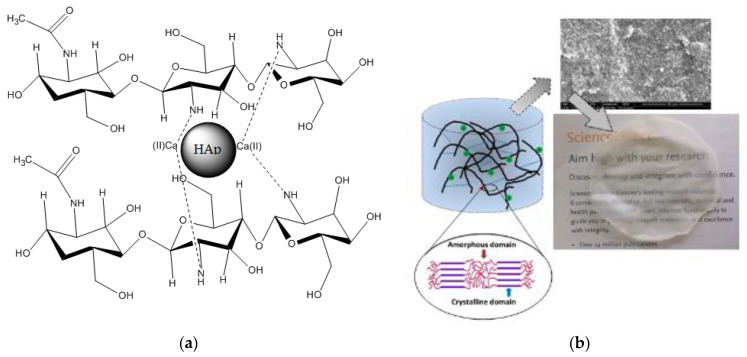
(**a**) Schematic model illustrations the relations between the NH groups of CTs and Ca of hydroxyapatite; (**b**) A SEM image, a photo of the obtained HAp/CTs membrane and a schematic diagram of membrane materials overlaying.

**Figure 8 polymers-14-04265-f008:**
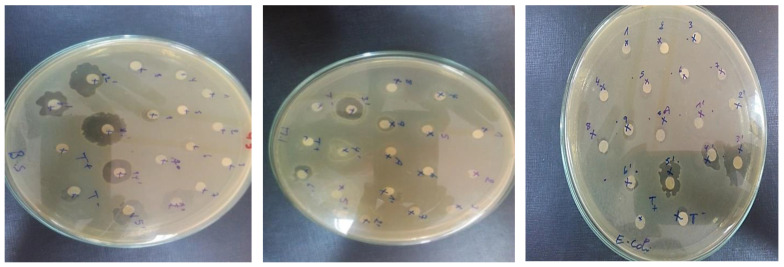
Sensitivity test in agar media.

**Figure 9 polymers-14-04265-f009:**
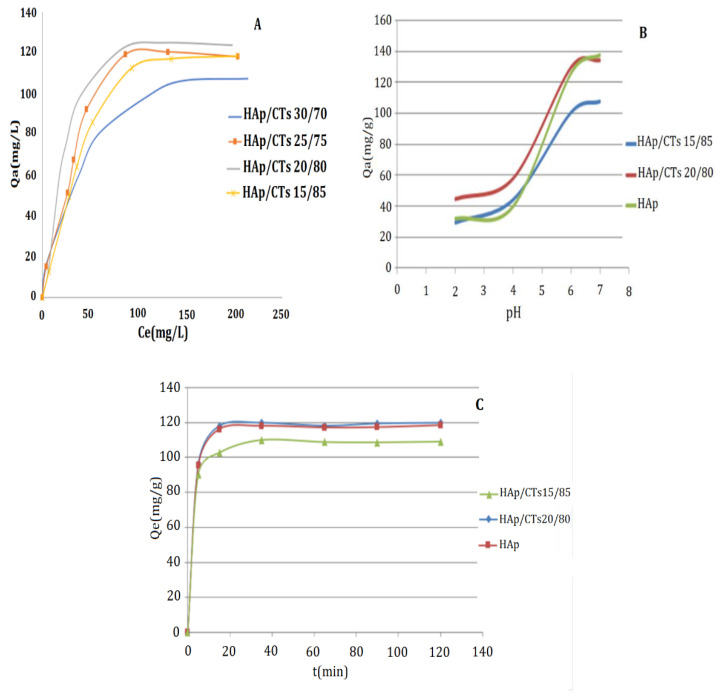
Effect of (**A**) initial concentration; (**B**) pH; and (**C**) contact time on the adsorption of Cd^2^ on HAp/CTs composites.

**Figure 10 polymers-14-04265-f010:**
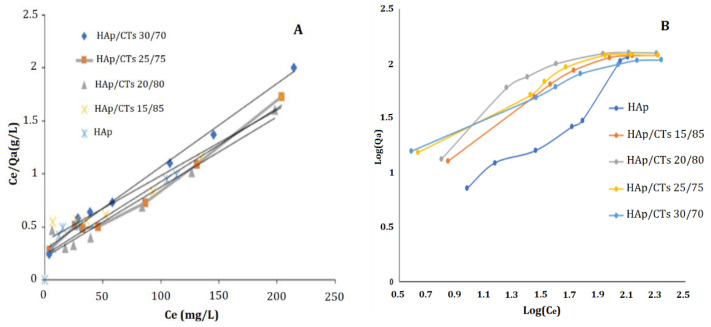
(**A**) Langmuir; and (**B**) Freundlich adsorption models for Cd^2+^ on HAp/CTs composites.

**Figure 11 polymers-14-04265-f011:**
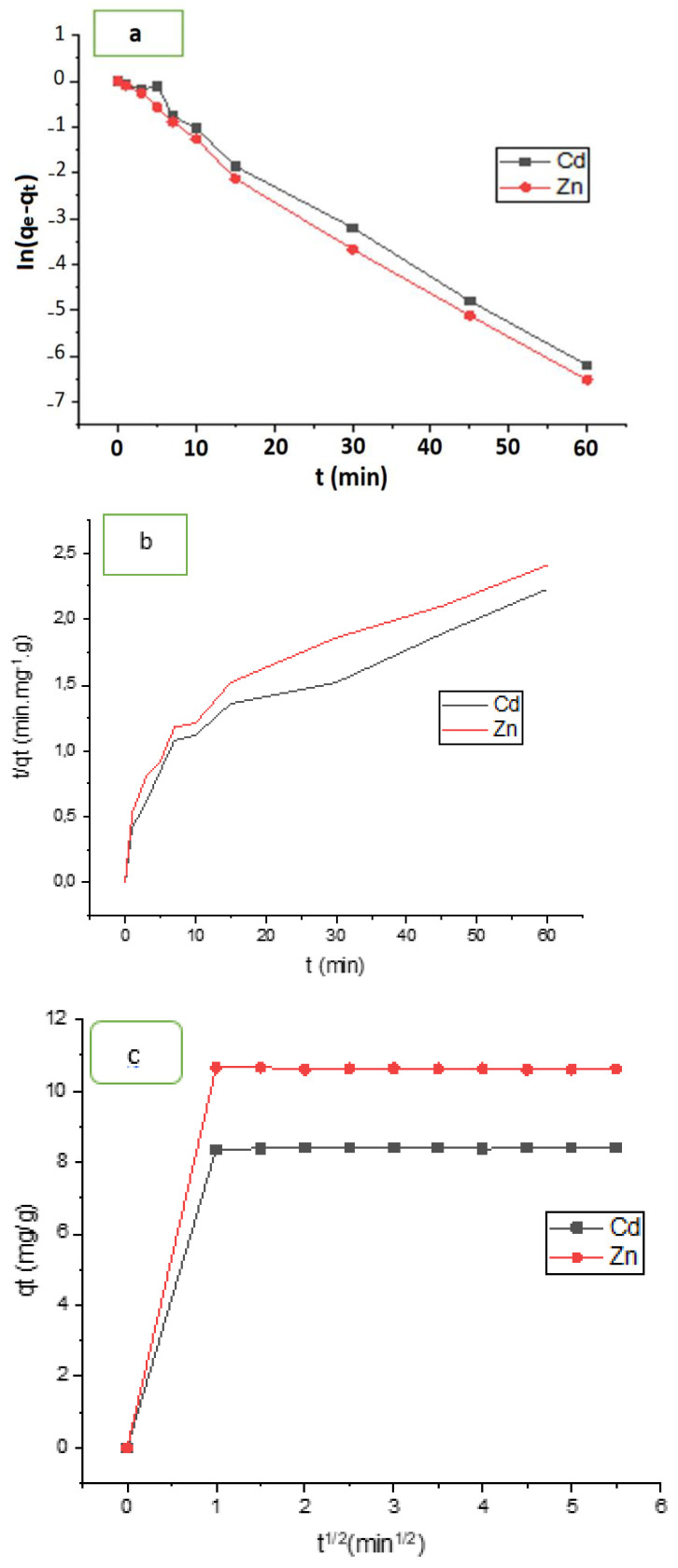
(**a**) Pseudo-first-order; (**b**) Pseudo-second-order; and (**c**) intra-particle diffusion models for Cd^2+^ and Zn^2+^.

**Figure 12 polymers-14-04265-f012:**
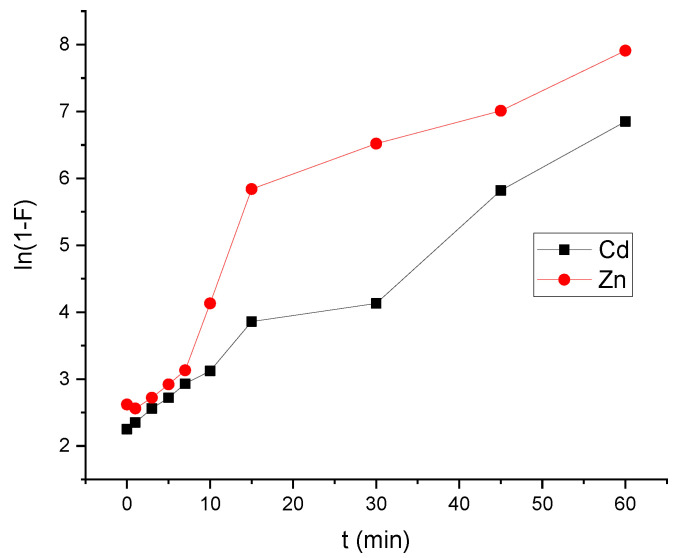
Liquid film diffusion process for the adsorption of HAp-based composites.

**Figure 13 polymers-14-04265-f013:**
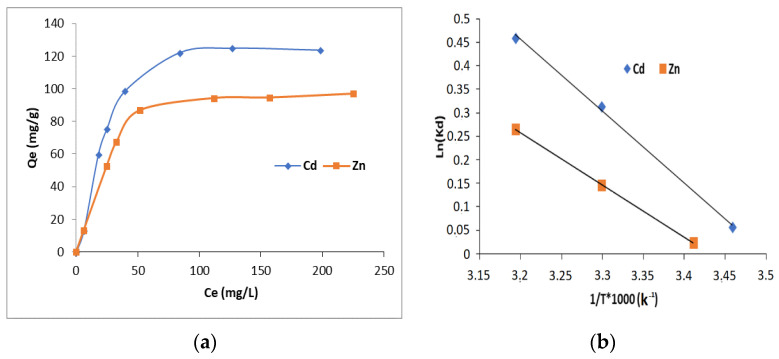
(**a**) Concentration effect on the adsorption of cadmium and zinc on HAp/CTs 20/80; (**b**) adsorption thermodynamics of Cd^2+^ and Zn^2+^ onto composite HAp/CTs 20/80.

**Figure 14 polymers-14-04265-f014:**
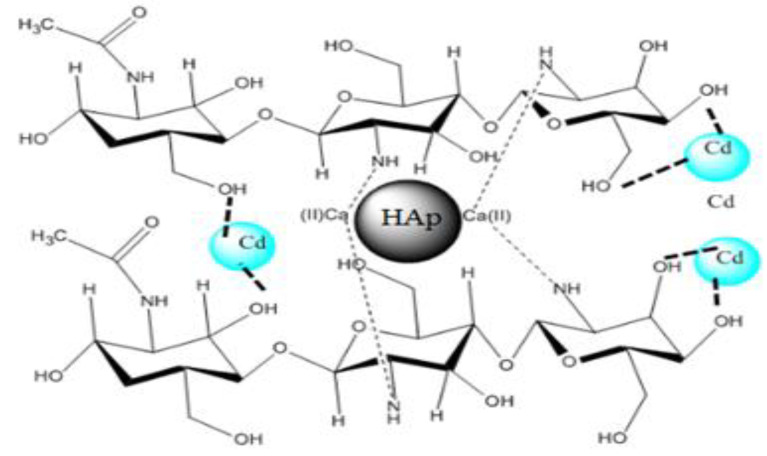
A 3D scheme for the possible interaction between composite HAp/CTs and Cd^+2^.

**Table 1 polymers-14-04265-t001:** The average size of HAp/CTs apatitic nanoparticles based on Scherrer’s formula.

Composite	Plan hkl	Dm (nm)
HAp/CTs15/85	002	20
	310	
HAp/CTs20/80	002	33
	310	
HAp/CTs25/75	002	36
	310	

**Table 2 polymers-14-04265-t002:** Parameters of Cd^2+^ adsorption isotherms on the composites.

Materials HAp/CTs	Langmuir Isotherm			Freundlich Isotherm		
	Q_m_ (mg/g)	K_L_ (L/mg)	R^2^	K_F_ (mg/g)	1/n	R^2^
**30/70**	128.21	2.74 × 10^−2^	0.9911	2.59	0.5008	0.9686
**25/75**	144.93	2.97 × 10^−2^	0.9773	2.50	0.5602	0.9207
**20/80**	151.52	3.04 × 10^−2^	0.9497	2.49	0.588	0.7858
**15/85**	161.29	1.70 × 10^−2^	0.9563	1.96	0.672	0.9073
**HApc**	169.49	1.19 × 10^−2^	0.9615	1.74	0701	0.9534

**Table 3 polymers-14-04265-t003:** Thermodynamic parameters of Cd^2+^ and Zn^2+^ adsorption on HAp/CTs 20/80 composite.

	T (°C)	T (K)	K_d_	ln(K_d_)	1/T	ΔG (kJ·mol^−1^)	ΔH° (kJ·mol^−1^)	ΔS° (k^−1^·J·mol^−1^)
**Cd^2+^**	16	289.15	1.068	0.058	0.0035	−0.1485	12.67	44.33
	30	303.15	1.369	0.314	0.0033	−0.7692		
	40	313.15	1.584	0.460	0,0032	−1.2126		
**Zn^2+^**	16	293.15	1.059	0.058	0.0035	−0.0564	9.237	31.0
	30	303.15	1.369	0.314	0.0033	−0.3734		
	40	313.15	1.584	0.460	0.0032	−0.6904		

**Table 4 polymers-14-04265-t004:** Different kinetic model parameters for the composite based of HAp/CTs.

Pseudo-First-Order Kinetic Model				
	**Q_e,exp_ (mg/g)**	**Q_e,cal_ (mg/g)**	**K_1_**	**R^2^**
Cd^2+^	23.67	3.26	1.136	0.914
Zn^2+^	25.72	2.75	1.723	0.912
**Pseudo-Second-Order Kinetic Model**				
Cd^2+^	23.67	21.83	1.326	0.985
Zn^2+^	25.72	23.78	2.23	0.991
**Intra-Particle Diffusion Model**				
		K_id_ (mg/g·min^1/2^)		R^2^
Cd^2+^		0.286		0.995
Zn^2+^		0.314		0.996
**Liquid Film Diffusion Model**				
		K_Fd_ (min^−1^)		R^2^
Cd^2+^		2.312		0.913
Zn^2+^		2.264		0.923

## Data Availability

Not applicable.

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
