# Peer review of "Chitosan-Hydroxyapatite Bio-Based Composite in Film Form: Synthesis and Application in Wastewater"

_polymers, 2022, doi:10.3390/polym14204265_

Round 1

Reviewer 1 Report

Abstract

line 19   "substantial physical force", rephrase it in more elaborative way

line 20  "Cd+2 "  should be Cd+2

line 22   same like line 20

I prefer to tail the abstract by sentence explaining potential prospects of your developed material. 

Keywords

"Biocompatibility" is ireleavant and non significant for your application, probably you meant "Biodegradability"  

Introduction

line 35-38  talk about biocomposites not composites, through light on lignocellulosic composites and their history in water remedation

I recommend citing those

"Current status of cellulosic and nanocellulosic materials for oil spill cleanup"

"Effect of environmental conditions on date palm fiber composites"

Last sentence in intro should be "The main objective of this article is....................................................."

=========================================

Author Response

Dear Reviewer 1

Many thanks for your valuable comments. We answered all of them and I am sure they made the manuscript looks better.

Best regards

Reviewer 1

Open Review

English language and style

( ) Extensive editing of English language and style required
(x) Moderate English changes required
( ) English language and style are fine/minor spell check required
( ) I don't feel qualified to judge about the English language and style

Yes

Can be improved

Must be improved

Not applicable

Does the introduction provide sufficient background and include all relevant references?

( )

(x)

( )

( )

Are all the cited references relevant to the research?

( )

( )

(x)

( )

Is the research design appropriate?

( )

(x)

( )

( )

Are the methods adequately described?

( )

(x)

( )

( )

Are the results clearly presented?

( )

(x)

( )

( )

Are the conclusions supported by the results?

( )

(x)

( )

( )

Comments and Suggestions for Authors

Abstract

line 19   "substantial physical force", rephrase it in more elaborative way   done

line 20  "Cd+2 "  should be Cd+2                                done

line 22   same like line 20                        done

I prefer to tail the abstract by sentence explaining potential prospects of your developed material.  done

Keywords

"Biocompatibility" is ireleavant and non-significant for your application, probably you meant "Biodegradability"              done

 Introduction

line 35-38 talk about biocomposites not composites, through light on lignocellulosic composites and their history in water remedation                       done

I recommend citing those                        done

"Current status of cellulosic and nanocellulosic materials for oil spill cleanup"                 done

"Effect of environmental conditions on date palm fiber composites"                done

Last sentence in intro should be "The main objective of this article is................ done..................................

Reviewer 2 Report

The manuscript "Chitosan-Hydroxyapatite Bio-Based Composite in film form: synthesis and application in Wastewater" deals with water purification from toxic metals. In particular, composite films constituted of hydroxyapatite, chitosan and glycerol were prepared by dissolution/recrystallization method to remove metals from water. Several analyses were performed; however, some improvements are required, as follows:

- Abstract. Add percentage of metals removal to this section.

- Introduction. The state of the art on water purification can be enlarged adding recent reviewes, as for instance: Somma et al., ChemEngineering2021, 5(3), 47; Syeda et al., Polymers, 2022, 14(12), 2341; Matei et al., Nanomaterials, 2022, 12(10), 1707; etc..

- R&D. SEM images shown in Figure 4 are not clear: improve quality. Film's morphology was closed and this kind of structure can reduce metals removal. Moreover, the observations of the authors cannot be deduced by SEM analysis; i.e., "The images show composite sheets with a homogenous and smooth surface indicating an excellent compatibility between composite components and a uniform distribution of chitosan and HAp in the composite. ". The statement: "The dense structure of the composite as observed by SEM micrographs illustrates the enhanced mechanical properties of the produced film.", is only an hypothesis that should be confirmed by mechanical tests of the samples.

- Use a better description for figure captions.

- Images shown in Figure 8 are not clear because of the perspective of the photo. 

- Improve English.

- Conclusions are a summary of the work. Rewrite in a more critical way, adding the main quantitative findings.

Author Response

Dear Reviewer 1

Many thanks for your valuable comments. We answered all of them and I am sure they made the manuscript looks better.

Best regards

Open Review

English language and style

( ) Extensive editing of English language and style required
( ) Moderate English changes required
( ) English language and style are fine/minor spell check required
(x) I don't feel qualified to judge about the English language and style

Yes

Can be improved

Must be improved

Not applicable

Does the introduction provide sufficient background and include all relevant references?

( )

(x)

( )

( )

Are all the cited references relevant to the research?

(x)

( )

( )

( )

Is the research design appropriate?

(x)

( )

( )

( )

Are the methods adequately described?

(x)

( )

( )

( )

Are the results clearly presented?

(x)

( )

( )

( )

Are the conclusions supported by the results?

(x)

( )

( )

( )

Comments and Suggestions for Authors

The manuscript "Chitosan-Hydroxyapatite Bio-Based Composite in film form: synthesis and application in Wastewater" deals with water purification from toxic metals. In particular, composite films constituted of hydroxyapatite, chitosan and glycerol were prepared by dissolution/recrystallization method to remove metals from water. Several analyses were performed; however, some improvements are required, as follows:

- Abstract. Add percentage of metals removal to this section. done

The adsorption studies showed that the maximum adsorption capacity of the HAp/CTs bio-composite membrane for Cd2+ and Zn2+ ions was in the order of cadmium (120 mg/g) > Zinc (90 mg/g) at an equilibrium time of 20 min and a temperature of 25°C.                

- Introduction. The state of the art on water purification can be enlarged adding recent reviewes, as for instance: Somma et al., ChemEngineering, 2021, 5(3), 47; Syeda et al., Polymers, 2022, 14(12), 2341; Matei et al., Nanomaterials, 2022, 12(10), 1707; etc..

- R&D. SEM images shown in Figure 4 are not clear: improve quality. Film's morphology was closed and this kind of structure can reduce metals removal. Moreover, the observations of the authors cannot be deduced by SEM analysis; i.e., "The images show composite sheets with a homogenous and smooth surface indicating an excellent compatibility between composite components and a uniform distribution of chitosan and HAp in the composite. ". The statement: "The dense structure of the composite as observed by SEM micrographs illustrates the enhanced mechanical properties of the produced film.", is only an hypothesis that should be confirmed by mechanical tests of the samples. done

- Use a better description for figure captions. done

- Images shown in Figure 8 are not clear because of the perspective of the photo.  done

- Improve English. done

- Conclusions are a summary of the work. Rewrite in a more critical way, adding the main quantitative findings.

Round 2

Reviewer 1 Report

Language is not super. However, I do recommend publication in the current form

Reviewer 2 Report

The authors performed all the modifications proposed by the Reviewer and improved the manuscript.